# 18-Hour Planar Scintigraphy Versus SPECT/CT for Sentinel Lymph Node Detection in Early-Stage Endometrial Cancer

**DOI:** 10.3390/cancers17182976

**Published:** 2025-09-11

**Authors:** Wiktor Szatkowski, Karolina Pniewska, Paweł Blecharz, Małgorzata Nowak-Jastrząb, Janusz Ryś, Tomasz Banaś, Renata Pacholczak-Madej, Emilia Krzywonos, Kamila Rawojć, Kamil Kisielewicz

**Affiliations:** 1Department of Gynecological Oncology, Maria Skłodowska-Curie National Research Institute, Kraków Branch, Garncarska 11, 31-115 Krakow, Poland; karolina.pniewska@krakow.nio.gov.pl (K.P.); pawel.blecharz@krakow.nio.gov.pl (P.B.); malgorzata.nowak@krakow.nio.gov.pl (M.N.-J.); renata.pacholczak@krakow.nio.gov.pl (R.P.-M.); 2Department of Pathology, Maria Skłodowska-Curie National Research Institute, Kraków Branch, Garncarska 11, 31-115 Krakow, Poland; janusz.rys@krakow.nio.gov.pl; 3Department of Radiotherapy, Maria Skłodowska-Curie National Research Institute, Kraków Branch, Garncarska 11, 31-115 Krakow, Poland; 4Department of Medical Physics, Maria Sklodowska-Curie National Research Institute of Oncology, Krakow Division, Garncarska 11, 31-115 Krakow, Poland; kamila.rawojc@dworska.pl (K.R.); kamil.kisielewicz@krakow.nio.gov.pl (K.K.)

**Keywords:** endometrial cancer, sentinel lymph node, planar scintigraphy, SPECT/CT, technetium-99m, lymphatic mapping, signal-to-noise ratio, contrast factor, nuclear medicine

## Abstract

Sentinel lymph node (SLN) mapping is a key step in the surgical staging of early-stage endometrial cancer (EC, FIGO I–II). While SPECT/CT is often considered superior to planar scintigraphy, it entails higher radiation exposure, greater costs, and limited accessibility. We compared three imaging protocols using technetium-99m (Tc-99m): early planar scintigraphy (30 min post-injection), SPECT/CT (1 h post-injection), and delayed planar scintigraphy at 18 h. The 18-h protocol achieved the highest SLN detection rate (94.4%) and bilateral detection rate (80.8%) in 125 patients, with complete intraoperative and histopathological confirmation. It was equally effective in obese patients (BMI ≥ 30, 94%). Image quality, assessed using the contrast factor (C-factor) and signal-to-noise ratio (SNR), supported these findings. Delayed planar scintigraphy is a highly effective, low-cost, and accessible method for SLN detection in early-stage EC, potentially eliminating the need for SPECT/CT. Further studies, including those involving artificial intelligence, may help to standardize this approach.

## 1. Introduction

Endometrial cancer (EC) is most often confined to the uterus, with lymph node metastases occurring infrequently [1]. Sentinel lymph node (SLN) biopsy is now recognized as the standard method for surgical staging in EC, offering a less invasive alternative to full lymphadenectomy while maintaining high diagnostic accuracy for metastasis detection [2,3]. This approach reduces the risk of complications such as lymphedema or lymphocyst formation, which is particularly important in obese patients—a common comorbidity in EC (74.4% of patients in our cohort had a BMI ≥ 30).

Various tracers are used for SLN mapping, including technetium-99m (Tc-99m), indocyanine green (ICG), and blue dyes (e.g., Patent Blue). Tc-99m is a widely used radiotracer due to its stability, deep tissue penetration, and long retention time in lymph nodes, as well as its applicability in both planar scintigraphy and hybrid SPECT/CT imaging [4,5,6]. Scintigraphy enables the assessment of SLN presence and its unilateral or bilateral location, and it helps to plan alternative strategies (e.g., dye usage) in cases of non-detection [7,8,9]. Although Tc-99m is highly effective in EC, its utility in other cancers (e.g., melanoma, head and neck cancer) confirms the versatility of this method [4,10].

The literature suggests the superiority of hybrid SPECT/CT imaging over planar scintigraphy, both in tumors with complex anatomy [10,11] and in gynecologic malignancies [12,13,14,15,16]. However, the effectiveness of sentinel lymph node (SLN) detection depends significantly on the timing of imaging. In most comparative studies, SPECT/CT was performed within so-called short protocols—that is, within three hours after Tc-99m injection, *which may not allow sufficient time for optimal tracer accumulation in SLNs* [12,13,14,15,16].

In our previous study [17], we demonstrated the high sensitivity of the 18 h planar scintigraphy protocol in a smaller cohort, with no cases of false uptake (“empty pockets”) and minimal background signal from the injection site. The 18-h timing was selected based on prior evidence suggesting optimal Tc-99m retention in SLNs [17,18] and practical considerations, allowing Tc-99m injection the day before and imaging on the morning of surgery (approximately 7:20 AM), with procedures performed 1–2 h later.

In the present study, we expanded our prior analysis [17] to a cohort of 125 patients with early-stage endometrial cancer (FIGO I–II), comparing three imaging protocols for sentinel lymph node (SLN) detection: 30-min planar scintigraphy, SPECT/CT at 1 h, and 18-h planar scintigraphy. We evaluated the learning curve effect on bilateral SLN detection and applied quantitative metrics (contrast factor (C-factor) ≥ 3.61, signal-to-noise ratio (SNR) ≥ 1.46) for objective image quality assessment.

The aim of the study was to evaluate whether delayed planar scintigraphy with Tc-99m is sufficiently sensitive and clinically effective (including positive and negative predictive values (PPV, NPV) to replace SPECT/CT in standard SLN detection in EC, thereby reducing radiation exposure, cost, and preoperative preparation time.

## 2. Materials and Methods

### 2.1. Study Design and Setting

This was a retrospective cohort analysis conducted at the Maria Sklodowska-Curie National Research Institute of Oncology, Krakow Branch, Poland, between December 2016 and April 2025. It included patients with endometrial cancer (EC) preoperatively classified as FIGO stage I–II (2009 classification), based on clinical and imaging assessments [19], and who underwent sentinel lymph node (SLN) mapping. The study protocol was approved by the Ethics Committee of the National Research Institute of Oncology (approval no. 10/2025). All data were anonymized, and patient consent was obtained in accordance with the “Institutional Review Board Statement” and “Informed Consent Statement” sections.

### 2.2. Inclusion and Exclusion Criteria

Eligible patients had histologically confirmed stage I–II endometrial cancer, had received no neoadjuvant therapy, and had complete clinical and pathological data. Exclusion criteria included advanced-stage disease (FIGO III–IV), age below 18 or above 85 years, and contraindications to surgical treatment. Of the 131 patients initially enrolled in the long planar scintigraphy protocol, 6 were excluded due to incomplete SPECT/CT imaging data, resulting in a final cohort of 125 patients for analysis.

### 2.3. Sentinel Lymph Node Mapping and Imaging

All patients underwent total hysterectomy, bilateral salpingo-oophorectomy or salpingectomy, and SLN mapping using technetium-99m (Tc-99m) human serum albumin colloid (NanoColl, GE Healthcare Limited, The Grove Centre, White Lion Road, Amersham, Bucks, UK). A dose of 120 MBq of Tc-99m was injected into the cervix at the 3 and 9 o’clock positions, both superficially (2–3 mm) and deeply (10–15 mm) into the cervical stroma using a 21G needle. Cervical injections were performed by three trained operators following a center-specific SOP. Inter-operator variability was mitigated by standard training and assessed indirectly via stable quantitative metrics.

Sentinel lymph nodes were defined as the first lymph nodes demonstrating radiotracer uptake in a given imaging protocol:Early planar scintigraphy—performed 30 min after injection (~113 MBq, acquisition time 600 s) using a Mediso, Budapest, Hungary, AnyScan SC gamma camera, field of view (FoV): 530 × 390 mm^2^, patient surface-to-detector distance: 3 cm.SPECT/CT—performed 1 h after injection (~107 MBq, acquisition time 600 s) using the Mediso AnyScan SPECT/CT system; the additional radiation dose from the CT component was estimated at 1–4 mSv [20], field of view (FoV): 530 × 390 mm^2^, patient surface-to-detector distance: 3 cm, attenuation correction: on.*Delayed planar scintigraphy was performed 18 h after injection (~15 MBq, acquisition time 600 s), on the day of surgery*, field of view (FoV): 530 × 390 mm^2^, patient surface-to-detector distance: 3cm. The median time from the 18-h planar acquisition to skin incision was 1.5 h (IQR: 1.0–2.0).

Intraoperative SLN localization was guided by the results of delayed planar scintigraphy and performed using a handheld gamma probe (Gamma Finder 2, World of Medicine Berlin, Germany). Identified SLNs were anatomically classified as obturator, internal iliac, external iliac, common iliac, or para-aortic, and ex vivo gamma probe confirmation was performed.

In two cases, hysterectomy was required before SLN mapping due to intense cervical activity; these patients were included in the denominator, and SLNs were subsequently identified after uterine removal. The overall detection rate was defined as the percentage of patients with at least one SLN visualized in a given imaging protocol. For the 18-h protocol, detection was additionally confirmed intraoperatively and histologically. Bilateral detection was defined as the identification of at least one SLN on each side of the pelvis.

### 2.4. Histopathological Analysis

Excised SLNs were fixed in formalin, sectioned into 2 mm slices, and stained with hematoxylin and eosin (H&E). Cytokeratin immunohistochemistry was used for metastasis detection and categorized as macrometastasis (>2 mm), micrometastasis (0.2–2 mm), or isolated tumor cells (≤0.2 mm).

### 2.5. Quantitative Metrics and Statistical Analysis

Scintigraphic image quality was assessed using the signal-to-noise ratio (SNR, threshold ≥ 1.46) and contrast factor (C-factor, threshold ≥ 3.61), in accordance with the methodology of Szatkowski et al. [17]. Regions of interest (ROIs) for SLNs were defined at 50% of the maximum pixel value [17], while background activity was measured in areas without tracer uptake [21]. The frequency of false-positive uptake (“empty pockets”) in fatty tissue was also recorded.

Statistical analyses were conducted using IBM SPSS Statistics v28.0 (IBM Corp., Armonk, NY, USA). Comparisons between imaging protocols (30-min planar, SPECT/CT, 18-h planar) were performed using ANOVA for quantitative metrics (SNR, C-factor). Paired *t*-tests were used for within-subject comparisons of SNR and C-factor. SLN detection and bilateral identification rates were analyzed using the chi-square test. A potential learning curve effect was evaluated by comparing bilateral detection rates in the last 17 consecutive cases with the overall cohort. A *p*-value of <0.05 was considered statistically significant.

### 2.6. SLN Evaluation Parameters

The following metrics were assessed:Detection sensitivity: The percentage of patients with at least one SLN identified and confirmed intraoperatively and histologically. This metric was based on delayed (18-h) scintigraphy, which was used for surgical planning.Bilateral detection: The proportion of cases with at least one SLN identified on each side of the pelvis.Positive predictive value (PPV) and negative predictive value (NPV): Calculated based on intraoperative and histopathological verification of SLNs identified on 18-h scintigraphic images.Quantitative metrics: Mean values and standard deviations of the signal-to-noise ratio (SNR) and contrast factor (C-factor), analyzed independently for each imaging protocol.

### 2.7. Ethical Approval and Informed Consent

The study was approved by the Ethics Committee of the Maria Sklodowska-Curie National Research Institute of Oncology, Krakow Branch, Poland (Approval No. 10/2025, date: 9 January 2025). Written informed consent was obtained from all subjects involved in the study, and all data were anonymized prior to analysis.

## 3. Results

The study cohort included 125 patients with endometrial cancer, with obesity (BMI ≥ 30) present in 74.4% of cases (Table 1). The 18-h planar scintigraphy protocol achieved the highest SLN detection sensitivity (94.4%, 118/125) compared to SPECT/CT (87.2%, 109/125; OR = 2.48; 95% CI: 0.98–6.27; *p* = 0.051) and the 30-min scintigraphy protocol (72%, 90/125) (Table 2, Figure 1). Bilateral SLN detection in the 18 h protocol was 80.8% (101/125) versus 73.6% (92/125) with SPECT/CT (OR = 1.51; 95% CI: 0.83–2.75; *p* = 0.16), with improvement to 88.2% (15/17) in the last 17 cases, suggesting a learning curve effect. The results of the 30-min and SPECT/CT protocols were not intraoperatively verified. Scintigraphic images in the 18 h protocol demonstrated improved SLN visualization and minimal background signal compared to the 30-min protocol (Figure 2 and Figure 3). No cases of so-called “empty pockets” (false uptake in fatty tissue) were observed in the 18 h protocol, as confirmed intraoperatively and histopathologically. Due to the lack of surgical verification in the 30-min and SPECT/CT groups, the presence or absence of empty pockets in these protocols cannot be definitively assessed.

The SLN detection sensitivity in patients with BMI ≥ 30 was 94% (87/93) for the 18 h protocol, 86% (80/93) for SPECT/CT, and 71% (66/93) for the 30-min protocol. All 118 cases with positive 18 h scintigraphy were confirmed intraoperatively and histopathologically as lymphatic tissue, resulting in a positive predictive value (PPV) of 100% (95% CI: 96.9–100%). In the seven cases where SLNs were not visualized with 18 h scintigraphy, the absence of intraoperative detection was also confirmed, suggesting a negative predictive value (NPV) of 100% (95% CI: 59.0–100%), though the limited number of negative cases reduces statistical power. No significant interaction between BMI category and imaging protocol was observed (*p* for interaction > 0.05), suggesting that the superiority of the 18-h protocol was consistent across BMI subgroups.

The total number of SLNs visualized in the 18 h protocol was 152, with a mean of 1.4 nodes per patient (range: 0–4) and median SLN per patient: 1.29. The most common anatomical locations included obturator nodes (40%), internal iliac (31%), external iliac (19%), common iliac (7%), and paraaortic nodes (3%).

Quantitative metrics confirmed high image quality in the 18 h protocol. The mean contrast ratio (C-factor) was 10.30 ± 1.22, slightly higher than for SPECT/CT (10.20 ± 1.30) and the 30-min protocol (10.02 ± 2.00). The mean signal-to-noise ratio (SNR) was 3.51 ± 1.2 for the 18 h protocol, compared to 4.22 ± 1.1 for SPECT/CT and 4.61 ± 1.1 for the 30-min protocol, reflecting the lower tracer activity used in the long protocol (15 MBq vs. 37.8 MBq [17]). Differences in C-factor values between protocols (18 h: 10.30 ± 1.22; SPECT/CT: 10.20 ± 1.30; 30 min: 10.02 ± 2.00) were not statistically significant (*p* > 0.05, paired *t*-test).

Differences in C-factor between protocols (18 h: 10.30 ± 1.22; SPECT/CT: 10.20 ± 1.30; 30 min: 10.02 ± 2.00) were not statistically significant (*p* > 0.05, paired *t*-test), but the 18 h protocol maintained superior clinical efficacy due to its higher detection sensitivity and PPV/NPV.

In two cases (1.6%) in the 30-min protocol, hysterectomy was required before SLN mapping due to signal interference near the cervix, which allowed for the successful identification of sentinel lymph nodes in that region.

## 4. Discussion

This study evaluated the effectiveness of planar scintigraphy performed at different time intervals after the administration of Tc-99m (120 MBq) compared to hybrid SPECT/CT for sentinel lymph node (SLN) detection in patients with endometrial cancer (EC). Three approaches were compared: planar scintigraphy performed 30 min after injection, hybrid SPECT/CT performed 1 h post-injection, and delayed planar scintigraphy performed 18 h post-injection (the so-called delayed protocol).

The highest SLN detection sensitivity was achieved with imaging performed 18 h after injection (94.40%, 118/125), outperforming both SPECT/CT (87.20%, 109/125; OR = 2.48; 95% CI: 0.98–6.27; *p* = 0.051) and early planar scintigraphy (72.00%, 90/125) (Table 2, Figure 1). Bilateral SLN detection with the 18-h imaging protocol was 80.80% (101/125) compared to 73.60% (92/125) with SPECT/CT (OR = 1.51; 95% CI: 0.83–2.75; *p* = 0.16). In the last 17 consecutive cases, bilateral detection increased to 88.2% (15/17), suggesting a learning curve effect associated with the standardized injection technique, operator experience, and multidisciplinary feedback.

Among the 152 SLNs detected using the delayed protocol, most were located in the obturator (40%), internal iliac (31%), external iliac (19%), common iliac (7%), and para-aortic (3%) regions.

Complete intraoperative and histopathological confirmation (100%) in the 18-h protocol and its high sensitivity in patients with BMI ≥ 30 (94%, 87/93) confirm its clinical utility, indicating that, in many cases, SPECT/CT may not provide additional benefits.

An ideal tracer for SLN mapping should exhibit rapid clearance from the injection site, high retention in SLNs, and minimal uptake in downstream nodes [22]. Our results are consistent with the SENTI-ENDO trial, which demonstrated 89% SLN detection sensitivity using Tc-99m in EC [23]. Due to its stability and precise intraoperative localization using a gamma probe, Tc-99m minimizes the risk of false-positive results, as evidenced by 100% intraoperative and histopathological concordance in the 18-h protocol.

In contrast to studies where SPECT/CT outperforms planar scintigraphy in short protocols (e.g., Kagoshima University, imaging within 1 h [12]; Navarro et al., short protocol [13]), our 18-h planar scintigraphy achieved higher sensitivity (94.40%) due to prolonged tracer accumulation in SLNs. Although performing SPECT/CT at 18 h could be considered, our data show that delayed planar scintigraphy alone achieved the highest SLN detection rate, with complete intraoperative and histopathological concordance in this cohort. Adding SPECT/CT at this time would introduce additional CT doses, costs, and logistical barriers without a clear expected clinical advantage in early-stage endometrial cancer, where lymphatic drainage is generally predictable. These findings support the use of delayed planar scintigraphy as a reliable and efficient approach in most patients. Similarly, Ogawa et al. reported 100% sensitivity and a 0% false-negative rate with Tc-99m phytate in cervical cancer [24], while Ballester et al. noted reduced sensitivity for same-day lymphoscintigraphy with next-day surgery in EC [18], supporting the importance of extended imaging timing. Togami et al. [12] and Bats et al. [7] reported higher false-negative rates with short protocols (23% and 16 h imaging, respectively), underscoring the critical role of timing.

Quantitative metrics such as SNR (3.51 ± 1.2 for 18 h; 4.22 ± 1.1 for SPECT/CT; 4.61 ± 1.1 for 30 min) and C-factor (10.30 ± 1.22 for 18 h; 10.20 ± 1.30 for SPECT/CT; 10.02 ± 2 for 30 min) enabled objective image quality assessment (Table 2). Although the SNR was lower in the 18-h protocol—due to lower tracer activity (15 MBq vs. 37.8 MBq [17])—the higher contrast coefficient (C-factor) indicates better SLN image contrast, supporting the detection effectiveness of this protocol.

Injection site signal interference occurred in only two patients (1.6%), requiring hysterectomy before SLN mapping. The minimal background signal in EC—unlike in cervical cancer (CC), where strong cervical uptake hinders imaging [25,26,27]—supports the adequacy of planar scintigraphy in EC. In CC, a meta-analysis conducted by Kraft and Havel showed 90% SPECT/CT sensitivity for SLNs in atypical locations [28], but, in EC, where SLN locations are generally predictable, the benefits of SPECT/CT appear limited, as noted by van der Ploeg et al. [29].

Our cohort included 74.4% of patients with BMI ≥ 30 (Table 1), and the 94% detection sensitivity in this group using the 18-h protocol exceeds the performance of ICG, where obesity is associated with a higher rate of SLN detection failure [30]. While SPECT/CT has shown improved accuracy in breast cancer patients with BMI > 30 (59% vs. 22% for planar scintigraphy [29]), the 18-h protocol in EC achieved comparable results without additional radiation from CT (1–4 mSv [31]). Importantly, no significant interaction between BMI category and imaging protocol was observed (*p* for interaction > 0.05), suggesting that the superiority of the 18-h protocol was consistent across BMI subgroups. These findings are highly relevant given the prevalence of obesity in EC; nonetheless, validation in non-obese populations and in different demographic groups is warranted.

A limitation of our study is that intraoperative and histopathological confirmation was available only for the 18-h planar protocol. By design, SLNs can be excised only once, making head-to-head histological verification of all imaging protocols within the same patient impossible outside a prospective randomized trial. Nevertheless, the complete intraoperative and histological concordance observed in the 18-h protocol strongly supports its reliability. Additional limitations include the small number of negative cases (n = 7), which reduces the precision of NPV estimates, and the wide confidence intervals for some odds ratios (e.g., 0.98–6.27 for sensitivity), reflecting the limited statistical power. These aspects underscore the need for larger multicenter studies to confirm our findings. Although shorter imaging times (e.g., 8–12 h) might also be effective, their performance could not be tested in our setting due to administrative constraints: Tc-99m was injected the day before, with imaging performed at approximately 7:20 AM on the day of surgery, followed by procedures one to two hours later. Further studies are required to evaluate the efficacy of intermediate-time protocols.

Future directions include the application of artificial intelligence (AI) in the analysis of scintigraphic images. Deep learning algorithms could automatically segment SLNs, improve detection accuracy in cases with high background signal, optimize Tc-99m dosing based on BMI, and predict signal interference risks, thereby supporting clinical decision making [17,21,32].

## 5. Conclusions

The 18-h planar scintigraphy with Tc-99m (120 MBq) proved to be a highly effective method for SLN detection in early-stage EC, achieving 94.4% sensitivity, 100% PPV/NPV, and 80.8% bilateral detection, with improvement to 88.2% in the last 17 cases, reflecting a learning curve effect. Its efficacy in patients with BMI ≥ 30 (94%) and minimal background interference suggests that delayed planar scintigraphy may reduce the routine need for SPECT/CT in many clinical scenarios, thereby lowering radiation exposure and costs. However, given the borderline statistical significance compared with SPECT/CT (*p* = 0.051) and the limited sample size, these findings should be interpreted with caution and validated in larger multicenter studies. Integration of artificial intelligence may further enhance the reproducibility and clinical utility of this protocol in the future.

## Figures and Tables

**Figure 1 cancers-17-02976-f001:**
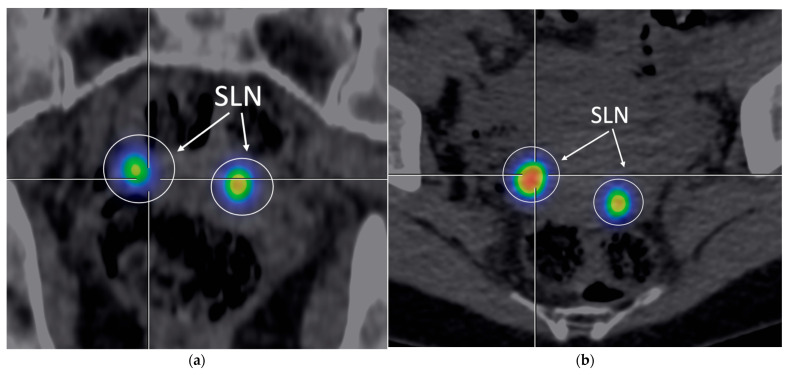
SPECT/CT. Representative image of SPECT/CT performed 1 h after Tc-99m injection (107 MBq), showing SLN uptake with moderate background signal, with a detection sensitivity of 87.20% (Table 2). (**a**) Frontal view; (**b**) transversal view.

**Figure 2 cancers-17-02976-f002:**
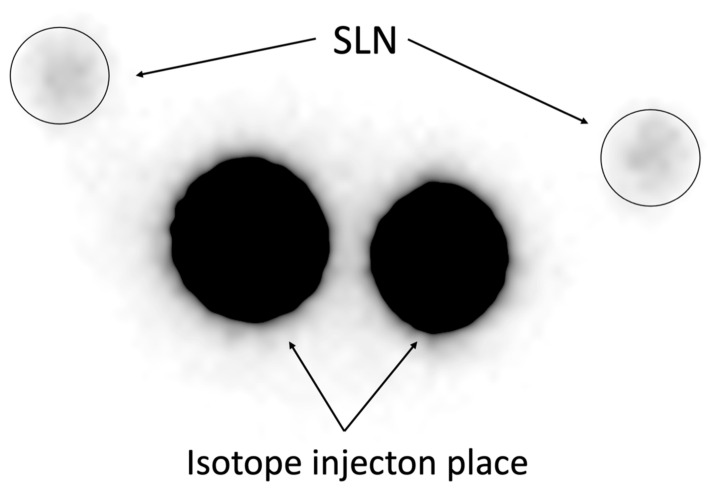
Image of 30-min planar scintigraphy. Representative image of planar scintigraphy performed 30 min after Tc-99m injection (113 MBq), showing SLN uptake with notable background signal interference, resulting in lower detection sensitivity (72.00%, Table 2).

**Figure 3 cancers-17-02976-f003:**
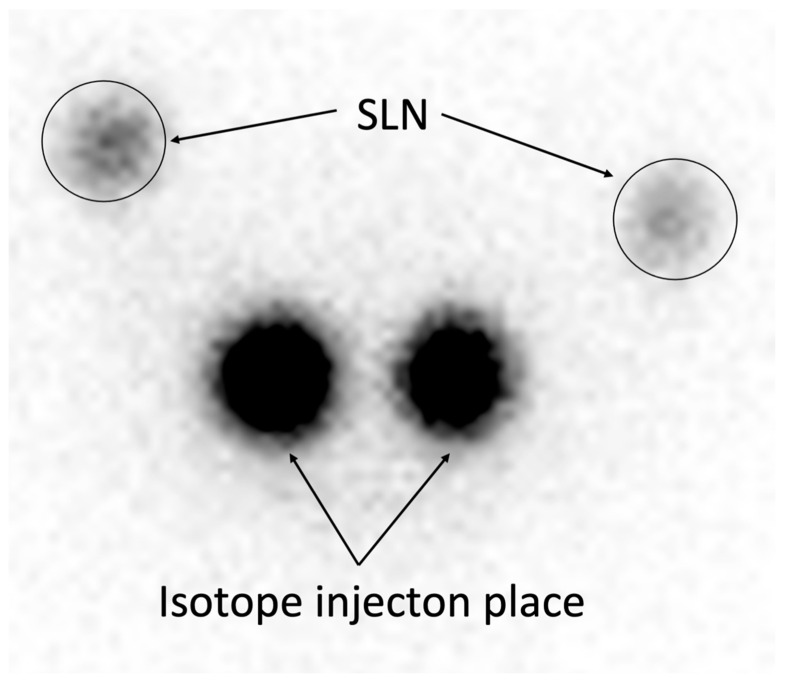
Eighteen-hour planar scintigraphy. Representative image of planar scintigraphy performed 18 h after Tc-99m injection (15 MBq), demonstrating clear SLN visualization with minimal background interference, achieving high detection sensitivity (94.40%) and 100% PPV/NPV (Table 2).

**Table 1 cancers-17-02976-t001:** Patient and tumor characteristics ^1^.

Feature	Characteristic	n	%
Age (years)	Mean ± SD	67.5 ± 9.7	–
Histologic Type	Endometrioid	109	87.20%
	Serous	10	8.00%
	Clear-Cell	6	4.80%
Lymphovascular Space Invasion (LVSI)	Present	17	13.60%
	Absent	108	86.40%
Myometrial Invasion	0%	19	15.20%
	<50%	74	59.20%
	>50%	32	25.60%
Lymphadenectomy	Performed	22	17.60%
	Not Performed	103	82.40%
FIGO Stage	IA	29	23.20%
	IB	57	45.60%
	II	16	12.80%
	IIIA	4	3.20%
	IIIB	2	1.60%
	IIIC1	13	10.40%
	IIIC2	4	3.20%
FIGO Grade	G1	60	48.00%
	G2	53	42.40%
	G3	12	9.60%
BMI Category	<25	8	6.40%
	25–29.9	19	15.20%
	≥30	93	74.40%
Total		125	100.00%

^1^ Legend: BMI categories: <25 (normal), 25–29.9 (overweight), ≥30 (obese). Lymphadenectomy: Refers to bilateral pelvic and/or paraaortic lymphadenectomy, performed in addition to sentinel lymph node (SLN) biopsy. All patients underwent SLN mapping, with at least one SLN detected intraoperatively in 118/125 patients using the 18-h planar scintigraphy protocol. n = 125 patients, representing the entire study cohort.

**Table 2 cancers-17-02976-t002:** Effectiveness of sentinel lymph node (SLN) detection in endometrial cancer by imaging protocol ^1^.

Parameter	30-Minute Planar Scintigraphy	SPECT/CT	18-Hour Planar Scintigraphy
Overall Detection Rate (%)	72.00% (90/125) *	87.20% (109/125) *	94.40% (118/125)
Bilateral Detection Rate (%)	60.00% (75/125) * 95% CI: 51.1–68.4%)	73.60% (92/125) * 95% CI: 65.1–80.7%)	80.80% (101/125) 95% CI: 72.9–87.3%)
Intraoperative Confirmation (%)	Not verified *	Not verified *	100.00%
Histopathological Confirmation (%)	Not verified *	Not verified *	100.00%
PPV (%)	Not verified *	Not verified *	100.00% (95% CI: 96.9–100.0)
NPV (%)	Not verified *	Not verified *	100.00% (95% CI: 59.0–100.0)
SNR (mean ± SD)	4.61 ± 1.10	4.22 ± 1.10	3.51 ± 1.20
C-factor (mean ± SD)	10.02 ± 2.00 (range: 3.46–21.26)	10.20 ± 1.30 (range: 3.48–21.5)	10.30 ± 1.22 (range: 3.61–21.27)
Sensitivity, BMI ≥ 30 (%)	71.00% (66/93) *	86.00% (80/93) *	94.00% (87/93)

^1^ Legend: * Data for 30-min planar scintigraphy and SPECT/CT are based solely on imaging results, without intraoperative or histopathological verification. In the final 17 consecutive cases, bilateral SLN detection in the 18-h planar scintigraphy group reached 88.20% (15/17, 95% CI: 65.7–97.4%), indicating a learning curve effect. The learning curve effect was assessed by comparing bilateral SLN detection in the last 17 consecutive cases (88.20%) with the entire cohort (80.80%) using the chi-square test. In two cases, hysterectomy was required to eliminate uterine signal interference. n = 125 patients per imaging protocol group; for BMI ≥ 30, n = 93 patients. SNR and C-factor are unitless metrics of image quality. Odds ratio (OR) for SLN detection sensitivity (18 h vs. SPECT/CT): OR = 2.48 (95% CI: 0.98–6.27); for bilateral detection rate: OR = 1.51 (95% CI: 0.83–2.75). *p*-value for SLN detection sensitivity (18 h vs. SPECT/CT): *p* = 0.051; for bilateral detection rate: *p* = 0.16 (chi-square test). Mean SLN per patient (18 h protocol): 1.4 (range: 0–4); median SLN per patient: 1.29.

## Data Availability

The original contributions presented in this study are included in the article. Further inquiries can be directed to the corresponding author(s).

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
