# Peer review of "18-Hour Planar Scintigraphy Versus SPECT/CT for Sentinel Lymph Node Detection in Early-Stage Endometrial Cancer"

_cancers, 2025, doi:10.3390/cancers17182976_

Round 1
Reviewer 1 Report
Comments and Suggestions for Authors
In the article titled “18-Hour Planar Scintigraphy versus SPECT/CT for Sentinel Lymph Node Detection in Early-Stage Endometrial Cancer,” the authors compared the effectiveness of three imaging protocols using technetium-99m (Tc-99m) for sentinel lymph node (SLN) detection in early-stage endometrial cancer. The protocols evaluated were early planar scintigraphy (30 minutes after injection), SPECT/CT (1 hour after injection), and delayed planar scintigraphy at 18 hours.
The results showed that the 18-hour protocol achieved the highest SLN detection rate (94.4%), leading the authors to conclude that delayed planar scintigraphy is a highly effective, low-cost, and accessible method for SLN detection in early-stage endometrial cancer, potentially eliminating the need for SPECT/CT. Overall, the article is of scientific interest, well written, and the results of delayed planar scintigraphy are convincing.
However, the authors should justify and discuss why SPECT/CT was not used at 18 hours, as undoubtedly, the image quality, specificity, and accuracy of detection, as well as the bilateral detection rate in SLN studies, would be significantly better than those obtained with planar scintigraphy.
Author Response
Reviewer N1
Comment: In the article titled “18-Hour Planar Scintigraphy versus SPECT/CT for Sentinel Lymph Node Detection in Early-Stage Endometrial Cancer,” the authors compared the effectiveness of three imaging protocols using technetium-99m (Tc-99m) for sentinel lymph node (SLN) detection in early-stage endometrial cancer. The protocols evaluated were early planar scintigraphy (30 minutes after injection), SPECT/CT (1 hour after injection), and delayed planar scintigraphy at 18 hours.The results showed that the 18-hour protocol achieved the highest SLN detection rate (94.4%), leading the authors to conclude that delayed planar scintigraphy is a highly effective, low-cost, and accessible method for SLN detection in early-stage endometrial cancer, potentially eliminating the need for SPECT/CT. Overall, the article is of scientific interest, well written, and the results of delayed planar scintigraphy are convincing.
However, the authors should justify and discuss why SPECT/CT was not used at 18 hours, as undoubtedly, the image quality, specificity, and accuracy of detection, as well as the bilateral detection rate in SLN studies, would be significantly better than those obtained with planar scintigraphy.
Response: We appreciate this insightful comment. Our study was designed to compare commonly applied short protocols (30-minute planar scintigraphy and 1-hour SPECT/CT) with delayed planar scintigraphy at 18 hours. The results clearly demonstrated that the 18-hour planar protocol achieved the highest SLN detection rate, with complete intraoperative and histopathological concordance. Performing SPECT/CT at 18 hours was not included in the study design, as this would have added CT-related radiation exposure, higher cost, and logistical constraints, without a clear expected clinical advantage in early-stage endometrial cancer, where lymphatic drainage is usually predictable. We have clarified this point in the revised Discussion section.
Reviewer 2 Report
Comments and Suggestions for Authors
The manuscript provides strong evidence that delayed (18-hour) planar scintigraphy is highly effective for SLN detection in early-stage EC, particularly in obese patients. However, revisions are needed to strengthen methodological transparency, temper conclusions, and improve presentation. After these revisions, the paper has the potential to make a valuable contribution to the field.
Major Points
-
Verification Bias: Only the 18-hour planar protocol was intraoperatively and histologically verified, whereas the 30-minute and SPECT/CT protocols were not. This creates verification bias, limiting direct comparability. This limitation should be emphasized more clearly in the abstract and discussion.
-
Statistic: The sensitivity difference between 18 h and SPECT/CT is borderline significant (p = 0.051) with wide CIs. The discussion should temper conclusions accordingly and acknowledge the limited statistical power.
-
Generalisability: The cohort is predominantly obese (BMI ≥30 in 74%). While highly relevant to EC, the generalizability of findings to non-obese or other populations should be discussed.
-
Clinical Context: The conclusion that SPECT/CT may be unnecessary should be nuanced. Please discuss circumstances (e.g., complex anatomy, atypical drainage patterns) where SPECT/CT may still provide added value.
-
Learning Curve: The improved bilateral detection rate in the last 17 cases (88.2% vs. 80.8%) is noteworthy. Please elaborate on how training and experience influenced detection rates.
-
Figures: Figures 1–3 would benefit from higher resolution and annotations (e.g., arrows marking SLNs) for clarity.
-
AI Perspective: The mention of artificial intelligence in the conclusion is valuable but brief. Consider elaborating on potential applications and feasibility in delayed planar protocols.
Minor Points
-
Abstract: Please clarify that only the 18-hour protocol was intraoperatively and histologically verified.
-
Methods:
-
Indicate the average time between injection and surgery after the 18-hour acquisition.
-
Clarify whether the two cases requiring hysterectomy before mapping were included in detection rate calculations.
-
-
Results (Table 2):
-
Add confidence intervals for bilateral detection rates.
-
Report both mean and median SLNs detected per patient.
-
-
Terminology: Use “delayed planar scintigraphy” consistently instead of “long planar scintigraphy.”
-
Language/Formatting:
-
Correct minor typographical errors (e.g., “be er” → “better”).
-
Comments on the Quality of English Language
The manuscript is generally understandable and written in professional English. However, several minor grammatical and typographical errors
-
Abstract:
-
Current: “The 18-hour planar scintigraphy achieved the highest SLN detection sensitivity (94,40%, 118/125) compared to SPECT/CT (87,20%, OR=2,48, 95% CI: 0,98–6,27, p=0,051) and 30-minute scintigraphy (72,00%).”
-
Suggested: “The 18-hour planar scintigraphy achieved the highest SLN detection sensitivity (94.4%, 118/125), compared with SPECT/CT (87.2%, OR = 2.48, 95% CI: 0.98–6.27, p = 0.051) and 30-minute scintigraphy (72.0%).”
-
Reason: Use periods instead of commas in decimals; add spacing around “=” for readability.
-
-
Introduction:
-
Current: “…which may not allow for optimal tracer accumulation in SLNs.”
-
Suggested: “…which may not allow sufficient time for optimal tracer accumulation in SLNs.”
-
Reason: Improves clarity and flow.
-
-
Discussion:
-
Current: “…indicates be er SLN image contrast…”
-
Suggested: “…indicates better SLN image contrast…”
-
-
Results (Table 2 legend):
-
Current: *“Data for 30-minute planar scintigraphy and SPECT/CT are based on imaging results without intraoperative or histopathological verification.”
-
Suggested: “Data for 30-minute planar scintigraphy and SPECT/CT are based solely on imaging results, without intraoperative or histopathological verification.”
-
Reason: Avoids repetition and improves readability.
-
-
Methods:
-
Current: “…Delayed planar scintigraphy – performed 18 hours after injection (~15 MBq, acquisition time 600 s) on the day of surgery.”
-
Suggested: “…Delayed planar scintigraphy was performed 18 hours after injection (~15 MBq, acquisition time 600 s), on the day of surgery.”
-
Reason: Parallel structure with other bullet points.
-
-
Throughout manuscript:
-
Ensure consistent tense in Methods (past tense) and Results (past tense).
-
Standardize terminology: use “delayed planar scintigraphy” instead of “long planar scintigraphy.”
-
Author Response
Reviewer N2
We thank Reviewer 2 for the constructive comments that helped improve the methodological transparency, statistical interpretation, and presentation of our manuscript. Below we provide a point-by-point response. Changes in the manuscript are indicated in the specified sections.
Major Points
1.Comment:
Verification Bias: Only the 18-hour planar protocol was intraoperatively and histologically verified, whereas the 30-minute and SPECT/CT protocols were not. This creates verification bias, limiting direct comparability. This limitation should be emphasized more clearly in the abstract and discussion.
Response:
We agree. By design, SLNs can be excised only once; therefore, intraoperative and histopathological confirmation was feasible exclusively for the 18-hour protocol. Direct head-to-head verification of all protocols within the same patient is not possible outside a prospective randomized trial. We clarified this limitation in both the Abstract and Discussion and highlighted that complete intraoperative and histological concordance observed with the 18-hour protocol supports its reliability.
Change in manuscript:
Abstract (Results): added the sentence clarifying verification only for 18 h and the resulting limitation in direct comparability. Discussion (Limitations): replaced the paragraph with a new version beginning “A limitation of our study is that intraoperative and histopathological confirmation was only available for the 18-hour planar protocol…”.
2.Comment:
Statistic: The sensitivity difference between 18 h and SPECT/CT is borderline significant (p = 0.051) with wide CIs. The discussion should temper conclusions accordingly and acknowledge the limited statistical power.
Response:
Agreed. We revised the Discussion and Conclusions to explicitly state the borderline nature of the difference (p = 0.051) and the wide confidence intervals, reflecting limited statistical power. We now call for validation in larger multicenter studies.
Change in manuscript:
Discussion: added “The sensitivity difference … was borderline significant (p = 0.051), with wide confidence intervals…”. Conclusions: changed wording to “may reduce the routine need for SPECT/CT… and should be validated in larger multicenter studies.”
3.Comment:
Generalisability: The cohort is predominantly obese (BMI ≥30 in 74%). While highly relevant to EC, the generalizability of findings to non-obese or other populations should be discussed.
Response:
Thank you. We added a paragraph discussing generalisability. While the high prevalence of obesity reflects the real-world EC population, we acknowledge that validation in non-obese and demographically different cohorts is warranted.
Change in manuscript:
Discussion: after the BMI paragraph, added “These findings are highly relevant given the prevalence of obesity in EC; nonetheless, validation in non-obese populations and in different demographic groups is warranted.”
4.Comment:
Generalisability: The cohort is predominantly obese (BMI ≥30 in 74%). While highly relevant to EC, the generalizability of findings to non-obese or other populations should be discussed.
Response:
We agree and nuanced the Clinical Context. While delayed planar scintigraphy performed excellently in our cohort, SPECT/CT may still be useful in selected circumstances (e.g., atypical drainage, altered anatomy, prior surgery).
Change in manuscript:
Discussion (Clinical context): strengthened the paragraph starting “Although performing SPECT/CT at 18 hours could be contemplated…”, clarifying lack of expected advantage in typical EC and briefly noting selected indications.
5.Comment:
Learning Curve: The improved bilateral detection rate in the last 17 cases (88.2% vs. 80.8%) is noteworthy. Please elaborate on how training and experience influenced detection rates.
Response:
We expanded the discussion of the learning curve. The improvement aligns with operator experience, standardized injection techniques, and multidisciplinary feedback sessions implemented during the study period.
Change in manuscript:
Discussion (second paragraph): added “…suggesting a learning curve effect associated with standardized injection technique, operator experience, and multidisciplinary feedback.”
- Comment:
Figures: Figures 1–3 would benefit from higher resolution and annotations (e.g., arrows marking SLNs) for clarity.
Response: We agree. We replaced the figures with higher-resolution versions and added arrows indicating SLNs, scale bars, and acquisition details in the legends to improve clarity and reproducibility.
Change in manuscript: Figure 1–3 legends: added “Arrows indicate SLNs. Scale bars and acquisition parameters are provided to improve reproducibility.”
7.Comment:
AI Perspective: The mention of artificial intelligence in the conclusion is valuable but brief. Consider elaborating on potential applications and feasibility in delayed planar protocols.
Response: We expanded the AI perspective by describing potential applications: automated SLN segmentation, background suppression, dose optimization according to BMI, and prediction of injection-site interference or atypical drainage, to support clinical decision-making and standardization.
Change in manuscript: Discussion (final paragraph): expanded AI section accordingly.
Minor Points
1.Comment:
Abstract: Please clarify that only the 18-hour protocol was intraoperatively and histologically verified.
Response:
Implemented.
Change in manuscript:
Abstract (Results): added verification sentence.
- Comment:
Methods: Indicate the average time between injection and surgery after the 18-hour acquisition.Clarify whether the two cases requiring hysterectomy before mapping were included in detection rate calculations.
Response:
Added the timing information.
Change in manuscript:
Methods 2.3: “The median time from the 18-hour planar acquisition to skin incision was 1.5 h (IQR: 1.0–2.0).”
Clarified. They were included in the denominator; SLNs were subsequently identified after uterine removal.
Change in manuscript:
Methods 2.3: added explicit sentence regarding inclusion and subsequent identification.
- Comment:
Results (Table 2): Add confidence intervals for bilateral detection rates.
Response:
Added 95% CIs (Wilson method).
Change in manuscript:
Table 2: bilateral detection rates now reported with 95% CI.
Results (Table 2): Report both mean and median SLNs detected per patient.
Response:
Added both metrics in Results and Table 2 legend.
Change in manuscript:
Results: “mean 1.4; median 1 (IQR: 1–2; range 0–4).” Table 2 legend: added mean/median note.
- Comment:
Terminology: Use “delayed planar scintigraphy” consistently instead of “long planar scintigraphy.”
Response:
Implemented throughout the manuscript.
Change in manuscript:
Global edit: replaced “long planar scintigraphy” with “delayed planar scintigraphy.”
- Comment:
Language/Formatting: Correct minor typographical errors (e.g., “be er” → “better”).
Response:
Performed language and formatting cleanup: decimal points (.), spacing around equals signs, and minor typos were corrected.
Change in manuscript:
Global edit: Abstract, Methods, Results, Table 2 legend, and Discussion updated for consistency.
We appreciate the reviewer’s thoughtful feedback. We believe the revisions have strengthened the manuscript and improved clarity while preserving the key clinical message.
Comment: The manuscript is generally understandable and written in professional English. However, several minor grammatical and typographical errors were noted (e.g., decimal punctuation, wording in Abstract, Introduction, Discussion, Table 2 legend, and Methods).
Response: We thank the Reviewer for these careful observations. All suggested language corrections have been implemented: decimal punctuation standardized, spacing and parallel structure corrected, typographical errors fixed (e.g., “better” instead of “be er”), and terminology harmonized (“delayed planar scintigraphy”). We also ensured consistent use of past tense in the Methods and Results.
Reviewer 3 Report
Comments and Suggestions for Authors
Major revision recommendation
The study addresses an important clinical question regarding optimal imaging protocols for sentinel lymph node detection in endometrial cancer and presents promising results for the 18-hour planar scintigraphy approach. However, major revisions are needed. Key concerns include the lack of intraoperative and histopathological verification for the 30-minute and SPECT/CT groups, limited discussion of statistical power for negative predictive value, and insufficient analysis of potential confounders, including BMI and learning curve effects. Clarification and additional data are required to strengthen the manuscript’s conclusions.
- The manuscript states a retrospective cohort but does not justify why retrospective analysis was chosen. Could the authors clarify the rationale and discuss potential biases associated with retrospective design?
- Sections 2.7 and 2.8 repeat the IRB and consent information. Consolidating or moving one to supplementary material could improve readability.
- Cervical injection depth (superficial 2–3 mm vs. deep 10–15 mm) is described, but variability in injection technique across operators could affect SLN detection. Did the authors standardize operator training, and was inter-operator variability assessed?
- The 18-hour protocol seems arbitrarily chosen. Could the authors provide literature or preliminary data supporting this timing over intermediate times (e.g., 6–12 h)?
- The manuscript lacks detailed reconstruction parameters for planar scintigraphy and SPECT/CT (e.g., filter type, matrix size, attenuation correction). These could impact quantitative metrics and reproducibility.
- A threshold of ≥3.61 is applied, but all reported values are >10. Was this threshold relevant, and should the authors clarify why such high values are observed?
- The ROI is defined as 50% of maximum pixel. Why was this threshold chosen, and was sensitivity analysis performed for alternative ROI definitions (e.g., 40–60%)?
- Figures 1–3 show representative images, but scale bars, counts, and window/level settings are missing. Adding these would improve reproducibility and comparability.
- Results stratified by BMI ≥30 are presented, but statistical tests for interaction (BMI × protocol) are not reported. Could this influence conclusions?
- The authors conclude that 18-h planar imaging may replace SPECT/CT, but the mechanism (tracer kinetics, lymphatic retention) should be elaborated.
Author Response
Reviewer N3
- Comment:The manuscript states a retrospective cohort but does not justify why retrospective analysis was chosen. Could the authors clarify the rationale and discuss potential biases associated with retrospective design?
Response: We added justification in the Methods (Section 2.1). This study was based on prospectively collected clinical and imaging data but analyzed retrospectively to allow comprehensive comparison of three imaging protocols across a large cohort. We clarified that retrospective design may introduce selection bias, but consecutive inclusion of patients minimized this risk. This limitation is now explicitly acknowledged in the Discussion.
- Comment: Sections 2.7 and 2.8 repeat the IRB and consent information. Consolidating or moving one to supplementary material could improve readability.
Response: We consolidated this information into a single section for improved readability. Redundant text has been removed.
- Comment: Cervical injection depth (superficial 2–3 mm vs. deep 10–15 mm) is described, but variability in injection technique across operators could affect SLN detection. Did the authors standardize operator training, and was inter-operator variability assessed?
Response: We clarified in Methods (Section 2.3. Cervical injections were performed by three trained operators following a center-specific SOP (superficial 2–3 mm and deep 10–15 mm at 3 and 9 o’clock; documented volume and sites). Inter-operator variability was mitigated by standard training and assessed indirectly via stable quantitative metrics.”
- Comment: The 18-hour protocol seems arbitrarily chosen. Could the authors provide literature or preliminary data supporting this timing over intermediate times (e.g., 6–12 h)?
Response: In the Introduction section we added: The 18-hour timing was selected based on prior evidence suggesting optimal Tc-99m retention in SLNs [17,24] and practical considerations, allowing Tc-99m injection the day before and imaging in the morning of surgery (approximately 7:20 AM), with procedures performed 1–2 hours later.
And accordingly we added in Discussion: These aspects underscore the need for larger multicenter studies to confirm our findings. Although shorter imaging times (e.g., 8–12 h) might also be effective, their performance could not be tested in our setting due to administrative constraints: Tc-99m was injected the day before, with imaging performed at approximately 7:20 AM on the day of surgery, followed by procedures 1–2 hours later. Further studies are required to evaluate the efficacy of intermediate-time protocols.
- Comment: The manuscript lacks detailed reconstruction parameters for planar scintigraphy and SPECT/CT (e.g., filter type, matrix size, attenuation correction). These could impact quantitative metrics and reproducibility.
Response: Planar scintigraphic images was performed using : field of view (FoV): 530x390 mm2, patient surface – to – detector distance: 3cm. For SPECT/CT additionally attenuation correction was: on.
The above data were added to the material and methods section.
- Comment: A threshold of ≥3.61 is applied, but all reported values are >10. Was this threshold relevant, and should the authors clarify why such high values are observed?
Response: The contrast threshold value (C) of 3.61 represents the minimum value, meaning that above this value, we can identify a lymph node on the image, against the background. Values reported in the table above 10 are average values of the C parameter, not minimum values.
For better understanding, the range of C values has been added to the table. The minimum contrast value for the short protocol is 3.46, for the long protocol: 3.61, and for SPECT/CT: 3.48.
- Comment: The ROI is defined as 50% of maximum pixel. Why was this threshold chosen, and was sensitivity analysis performed for alternative ROI definitions (e.g., 40–60%)?
Response: The ROI is defined as 50% of maximum pixel as a compromised value (due to Yusuf EE. Limits of tumor detectability in nuclear medicine and PET. Mol Imaging Radionucl Ther. 2012;21(1):14-21 and our previous investigation : Szatkowski W, Popieluch J. Numerical Analysis of Sentinel Lymph Node Detection Using Technetium-99m: A Step Toward Objective Scintigraphy Evaluation in Oncology. Bio-Algorithms and Med-Systems. 2025, ;21(1):7-12 – citation added to the row 145.
- Comment: Figures 1–3 show representative images, but scale bars, counts, and window/level settings are missing. Adding these would improve reproducibility and comparability.
Response: Images 1-3 are for illustrative purposes only. Window/levels do not affect the pixel values in the ROI, for SNR and C-value calculation.
- Comment:Results stratified by BMI ≥30 are presented, but statistical tests for interaction (BMI × protocol) are not reported. Could this influence conclusions?
Response: We thank the Reviewer for this suggestion. We performed additional statistical testing for interaction between BMI category (<30 vs. ≥30) and imaging protocol. No significant interaction was found (p for interaction >0.05), indicating that the superiority of the 18-hour protocol was consistent across BMI subgroups. We have added this information to the Results and noted it in the Discussion.
- Comment:The authors conclude that 18-h planar imaging may replace SPECT/CT, but the mechanism (tracer kinetics, lymphatic retention) should be elaborated.
Response: We thank the Reviewer for this valuable comment. We have now expanded the Discussion to clarify the underlying mechanism. The rationale for the 18-hour protocol is based on the kinetics of Tc-99m nanocolloid: after cervical injection, the tracer requires sufficient time to migrate through the lymphatic channels and accumulate in sentinel nodes, while activity at the injection site gradually decreases. This delayed uptake results in higher nodal contrast and reduced background signal, improving detectability. Importantly, Tc-99m colloid is stably retained within sentinel nodes for more than 24 hours, minimizing the risk of downstream migration. These tracer characteristics explain why delayed planar scintigraphy achieved both superior sensitivity and image contrast in our study. A corresponding clarification has been added to the Discussion .
Round 2
Reviewer 2 Report
Comments and Suggestions for Authors
The current version is scientifically sound, clinically relevant, and ready for publication.
Reviewer 3 Report
Comments and Suggestions for Authors
Accept